# Array-Designed Triboelectric Nanogenerator for Healthcare Diagnostics: Current Progress and Future Perspectives

Zequan Zhao [1], Qiliang Zhu [1], Yifei Wang [1], Muhammad Shoaib [1], Xia Cao [2,3,*] and Ning Wang [1,2,*]

[1]  Center for Green Innovation, School of Mathematics and Physics, University of Science and Technology Beijing, Beijing 100083, China; m202110789@xs.ustb.edu.cn (Z.Z.); d202110424@xs.ustb.edu.cn (Q.Z.); d202210430@xs.ustb.edu.cn (Y.W.); muhammadshoaib123491@yahoo.com (M.S.)
[2]  Beijing Institute of Nanoenergy and Nanosystems, Chinese Academy of Sciences, Beijing 100083, China
[3]  School of Chemistry and Biological Engineering, University of Science and Technology Beijing, Beijing 100083, China
*   Correspondence: caoxia@ustb.edu.cn (X.C.); wangning@ustb.edu.cn (N.W.)

**Abstract:** Array-designed triboelectric nanogenerators (AD-TENGs) have firmly established themselves as state-of-the-art technologies for adeptly converting mechanical interactions into electrical signals. Central to the AD-TENG's prowess is its inherent modularity and the multifaceted, grid-like design that pave the way to robust and adaptable detection platforms for wearables and real-time health monitoring systems. In this review, we aim to elucidate the quintessential role of array design in AD-TENGs for healthcare detection, emphasizing its ability to heighten sensitivity, spatial resolution, and dynamic monitoring while ensuring redundancy and simultaneous multi-detection. We begin from the fundamental aspects, such as working principles and design basis, then venture into methodologies for optimizing AD-TENGs that ensure the capture of intricate physiological changes, from nuanced muscle movements to sensitive electronic skin. After this, our exploration extends to the possible cutting-edge electronic systems that are built with specific advantages in filtering noise, magnifying signal-to-noise ratios, and interpreting complex real-time datasets on the basis of AD-TENGs. Culminating our discourse, we highlight the challenges and prospective pathways in the evolution of array-designed AD-TENGs, stressing the necessity to refine their sensitivity, adaptability, and reliability to perfectly align with the exacting demands of contemporary healthcare diagnostics.

**Keywords:** triboelectric nanogenerators; array design; medical application; healthcare diagnostics

## 1. Introduction

Since their advent in 2012, TENGs have set a new benchmark in energy harvesting and active sensing, establishing their presence in a myriad of domains including green energy, molecular detection, healthcare, and gesture recognition [1–6]. Their dual functionality as energy harvesters and intelligent sensors positions TENGs as a promising solution for environmentally friendly and personalized healthcare. The array design in TENGs has emerged as a pivotal aspect in crafting TENG-based biosensors that expand their feasibility in health monitoring, environmental sensing, and point-of-care diagnostics by enhancing their sensitivity, adaptability, and spatial resolution [7–13].

However, despite the remarkable advancements in this field, challenges persist concerning power output, device stability, biocompatibility, and integration with other cutting-edge technologies such as flexible electronics and advanced data processing systems [14–21]. In recent years, research directions have been focused on addressing these limitations and exploring new pathways for amplifying the performance and capabilities of these adaptable devices, and AD-TENGs are distinctively recognized because their matrix configuration nurtures a diverse approach towards detection and energy harvesting [22–25]. This grid-like framework inherently houses essential qualities such as enhanced sensitivity, superior spatial resolution, redundancy, and the ability for concurrent multi-detection, elements

vital for dynamic monitoring functionalities [26–29]. Moreover, AD-TENGs are adept at registering subtle physiological alterations, thereby holding immense potential to steer the course of advancements in healthcare and wearable technology [30–36]. The overarching goal of the current research is to hone and maximize these attributes, promoting the evolution of robust, efficient, and precise data collection systems.

Compared to piezoelectric, electromagnetic, and thermoelectric, array-designed triboelectric nanogenerators (AD-TENGs) stand out due to several key advantages [37–43]. First, they exhibit high efficiency and energy output, particularly because their array design enhances power generation efficiency. Secondly, their modularity and scalability make them versatile for applications ranging from small-scale wearables to larger energy systems. AD-TENGs are also notable for their flexibility and adaptability, conforming easily to different shapes and surfaces, which is crucial for wearable healthcare devices [44]. The array configuration enhances both sensitivity and spatial resolution, allowing for the detection of subtle physiological changes, essential in healthcare diagnostics. Additionally, AD-TENGs offer redundancy and multi-detection capabilities, ensuring reliable performance even if individual units fail. In terms of fabrication, they are generally more cost-effective and simpler to produce than other nanogenerators, using low-cost materials and straightforward processes. Environmentally, AD-TENGs are favorable due to their reliance on non-toxic materials, making them more sustainable. Lastly, their versatility in energy harvesting from various mechanical movements, including human motion and natural elements like wind and water, makes them highly adaptable for diverse applications [45–48]. These attributes collectively position AD-TENGs as highly effective and versatile for a range of applications, particularly in healthcare diagnostics and wearable technology.

In this review, we aspire to present an in-depth analysis of the recent advancements and groundbreaking strategies in the design of AD-TENGs for simultaneous energy harvesting and medical utilities (Figure 1). We begin from the fundamental aspects by introducing the basis mechanism and working modes of AD-TENGs, then detail the latest advancements in the application of AD-TENGs in health monitoring and biosensors to highlight the versatility and potential (Table 1). After that, we explore various strategies for incorporating array designs into TENG-based biosensors that foster the development of innovative and customizable devices for contemporary healthcare diagnostics. To shed light on the incorporation of cutting-edge materials and their role in augmenting the efficiency and adaptability of AD-TENG-based devices across diverse sectors, we showcase the accomplishments of AD-TENGs in realms such as motion sensing and electronic skin, alongside the wide-ranging medical applications and the adoption of array structures in textile TENG. Furthermore, we examine the potential repercussions of these progressions on the frontier of novel energy solutions and state-of-the-art medical apparatuses, ultimately ushering in a fresh epoch of sustainable and individualized healthcare diagnostics solutions.

**Table 1.** Summary of AD-TENG.

| Data | Size | Energy Sources | Outputs | Applications | Working Modes |
|---|---|---|---|---|---|
| 2022 [49] | None | Vibration | 16.96 W m$^{-3}$ | Wave Energy Collection | Lateral sliding |
| 2022 [50] | 5 cm × 5 cm | Movement | 26 mW | Gait Recognition | Contact–separation |
| 2023 [51] | None | Movement | 6 nA. | Finger Bending Sensing | Contact–separation |
| 2022 [52] | 2 cm × 2 cm | Movement | 3 μA | Body Motion Sensing | Contact–separation |
| 2022 [53] | None | Vibration | 85 V | Energy Collection | Contact–separation |
| 2023 [54] | 7.5 cm × 7.5 cm | Vibration | 0.11 V/kPa | Pressure Sensing | Contact–separation |
| 2023 [55] | 1 cm × 1 cm | Movement | 15 nA | Tactile Sensing | Contact–separation |
| 2021 [56] | 2 cm × 2 cm | Vibration | 51.2 V | Body Motion Sensing | Contact–separation |
| 2022 [57] | 2.8 cm × 3 cm | Vibration | 200.93 mW/m$^2$ | Energy Collection | Contact–separation |
| 2020 [58] | 8 cm × 8 cm | Vibration | 7531 μW/m$^2$ | Energy Collection | Contact–separation |
| 2022 [59] | 4 cm × 4 cm | Movement | 469 mW/m$^2$ | Energy Collection | Contact–separation |
| 2022 [60] | None | Movement | 52 V | Energy Collection | Contact–separation |

**Table 1.** *Cont.*

| Data | Size | Energy Sources | Outputs | Applications | Working Modes |
|------|------|----------------|---------|--------------|---------------|
| 2022 [61] | 8 cm × 8 cm | Movement | 138.55 mW/m$^2$ | Body Motion Sensing | Contact–separation |
| 2023 [62] | 2 cm × 2 cm | Movement | 48 V | Body Energy Collection | Contact–separation |
| 2021 [63] | 3 cm × 3 cm | Vibration | 12 μA | Energy Collection | Contact–separation |
| 2020 [64] | None | Movement | 26.9 μA | Energy Collection | Contact–separation |
| 2022 [65] | None | Movement | 1.25 mW/m$^2$ | Dangerous Motion Sensing | Contact–separation |
| 2022 [66] | 2 cm × 2 cm | Vibration | 64 V | Sleep State Sensing | Contact–separation |
| 2022 [67] | None | Vibration | 230 V | Sterilization | Contact–separation |

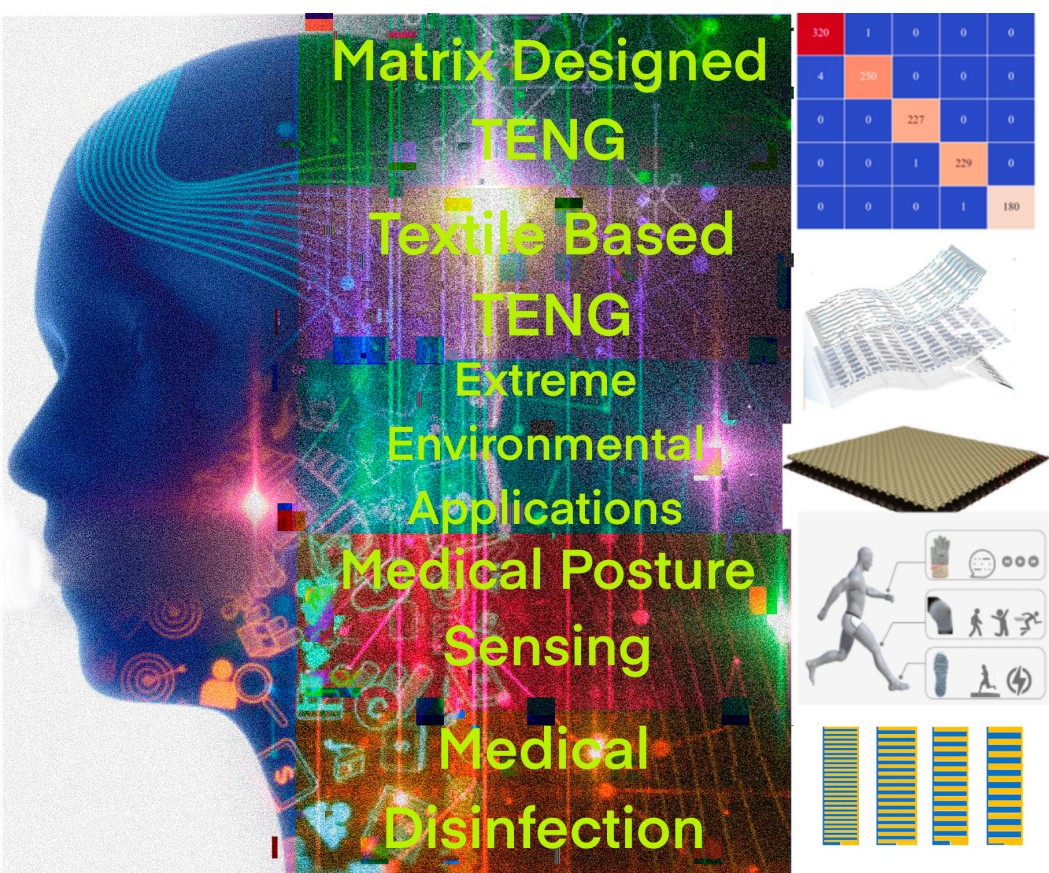

**Figure 1.** The comprehensive applications of array-designed TENG.

## 2. Array Configuration Amplifies TENG Energy Collection

### 2.1. The Principle TENG

TENGs, similar to TENGS with other various designs, capitalize on the triboelectric effect coupled with electrostatic induction to transform biomechanical energy into usable electrical energy [68–71]. This conversion process initiates when materials possessing distinct electronegativities come into contact, facilitating the transfer of electrons between them. Subsequently, as these materials part ways, an electrostatic induction initiates, steering the flow of electrons towards the external load and thereby engendering an alternating current that sustains through repeated cycles of contact and separation. In principle, TENGs predominantly exhibit four core modes of operation, which are delineated below and illustrated in Figure 2a.

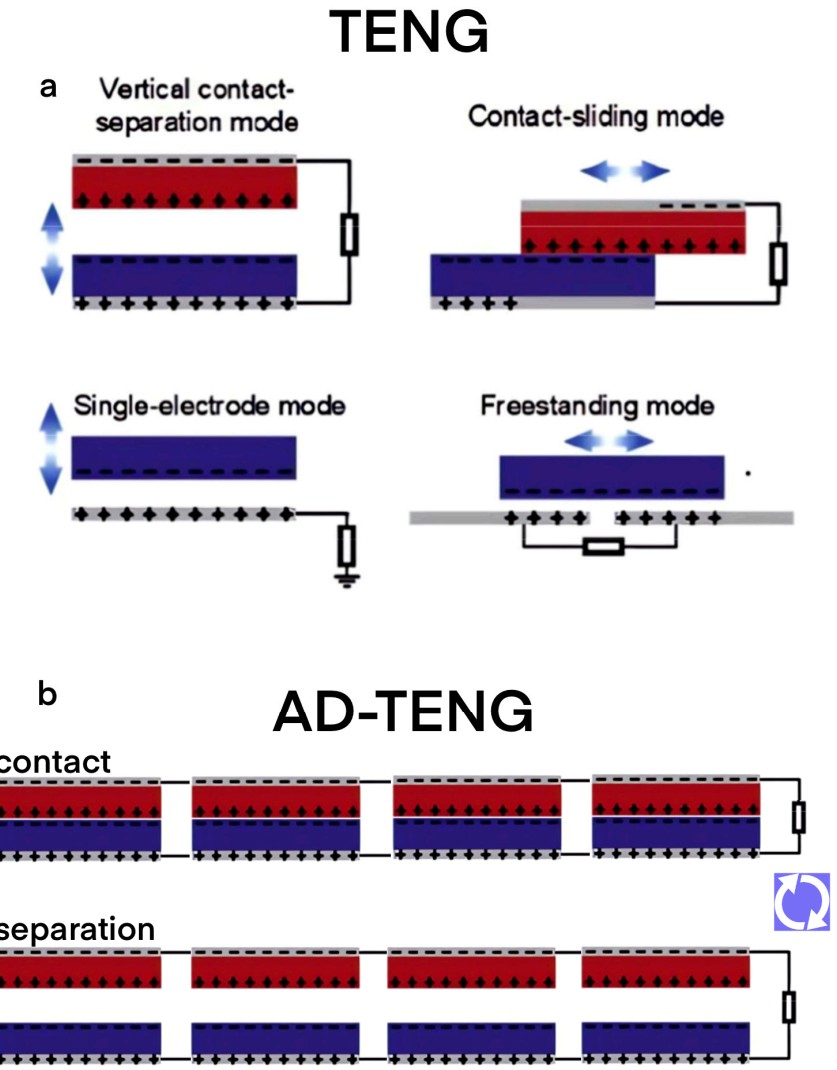

**Figure 2.** (**a**) Overview of TENG modes. (**b**) Overview of AD-TENG modes. 2023 MDPI [72].

Vertical Contact–Separation Mode: Within this modality, two triboelectric materials carrying opposing electrical charges are stationed in close proximity, undergoing periodic cycles of contact and separation along a vertical trajectory. This contact fosters the generation of triboelectric charges at the mutual interface. As the materials disengage, a redistribution of these charges occurs, fostering an electrical potential difference that instigates the flow of electrons through an external load, thereby generating electric power [73,74].

Lateral Sliding Mode: This mode witnesses two triboelectric materials, holding opposite charges, engaging in a horizontal sliding motion against each other. This relative movement, parallel to their interface, leads to a dynamic alteration in the overlapping area, thereby creating an electric potential gradient. This gradient acts as a catalyst for the flow of electrons through an external circuit, thereby generating electricity [75,76].

Single-Electrode Mode: In this scenario, one of the materials boasts an attached electrode, whereas the counterpart remains electrically isolated. This isolated component undergoes periodic cycles of contact and separation with the material having the attached electrode. This interaction engenders triboelectric charges at the interface, inducing a flow of electrons through the connected ground electrode and the singular electrode, thereby producing electrical power [77,78].

Freestanding Triboelectric Layer Mode: Here, a standalone triboelectric layer, flanked by electrodes on either side, exhibits opposing triboelectric charges on its two facets. This layer undergoes a sequence of contact and separation with the electrodes, inducing a defor-

mation in the layer. This mechanical action facilitates the generation of triboelectric charges at the interfaces between the freestanding layer and the electrodes. The induced electric potential difference propels the flow of electrons through an external circuit, culminating in the generation of electrical energy [79,80].

### 2.2. Design and Optimization of Array-Designed TENG

In an AD-TENG system, the integration of multiple TENG units into a singular array leads to a significant increase in total energy output (Figure 2b). Each unit contributes individually to the energy harvesting process, resulting in a cumulative effect that substantially elevates the overall power generation. This synergistic operation is a cornerstone in the design philosophy of AD-TENGs, providing a robust framework for efficient energy conversion.

Moreover, the array configuration ensures a uniform distribution of mechanical stress across the TENG units. This uniformity is crucial for the consistent activation of all units, thereby optimizing the energy harvesting efficiency. It also plays a key role in enhancing the durability and reliability of the system. The redundancy afforded by multiple units ensures that the failure of any single unit does not critically impair the overall performance, thereby extending the operational lifespan of the AD-TENG.

Flexibility in design is another hallmark of AD-TENGs. The array can be tailored in various shapes and sizes, making it highly adaptable to specific requirements, especially in applications like wearable technology and biomedical devices. This customizable nature of AD-TENGs opens avenues for their incorporation into a multitude of platforms, ranging from small-scale electronic devices to larger energy harvesting systems.

The scalability of AD-TENG arrays is an essential feature, allowing for adjustments in size and configuration to meet the desired power output for different applications. This scalability, coupled with the ease of integration with electronic circuitry, facilitates sophisticated control and optimization of harvested energy, which is particularly beneficial in applications requiring precise energy management, such as smart sensors and IoT devices.

Furthermore, the array design presents numerous opportunities for optimization in material selection, structural design, and operational modes. Tailoring each unit within the array to specific operational conditions can significantly enhance the overall performance of the AD-TENG. Additionally, in applications requiring high spatial resolution, such as sensing and health monitoring, the array design allows each unit to act as an independent sensor, providing detailed information about mechanical interactions or physiological parameters.

### 2.3. Improving Energy Collection Efficiency

In the realm of energy harvesting, the optimization of array structures to amplify both energy collection efficiency and stability stands as a focal point of innovation [58,81–83]. A notable breakthrough in this area is epitomized in Han et al.'s research, which unveils a remarkable stride in harmonizing efficiency and enduring performance through sophisticated array configurations [49]. In this study, a hybrid TENG was conceptualized and realized to exploit the intricate dynamics of ultra-low-frequency wave energy—a sector traditionally marked by its intricate patterns and directional unpredictability. The three strategically aligned TENGs significantly amplify both space utilization and volume power density, while ensuring a continuous and stable energy collection ability compared to other devices (Figure 3a). This accomplishment, evidenced by peak volume power densities of 2.02 and 16.96 W m$^{-3}$ for F-TENG and H-EMG, respectively, at a 1.4 Hz stimulation frequency, delineates a significant advancement in the creation of self-powered intelligent marine monitoring systems and a spectrum of energy harvesting applications within smart city frameworks.

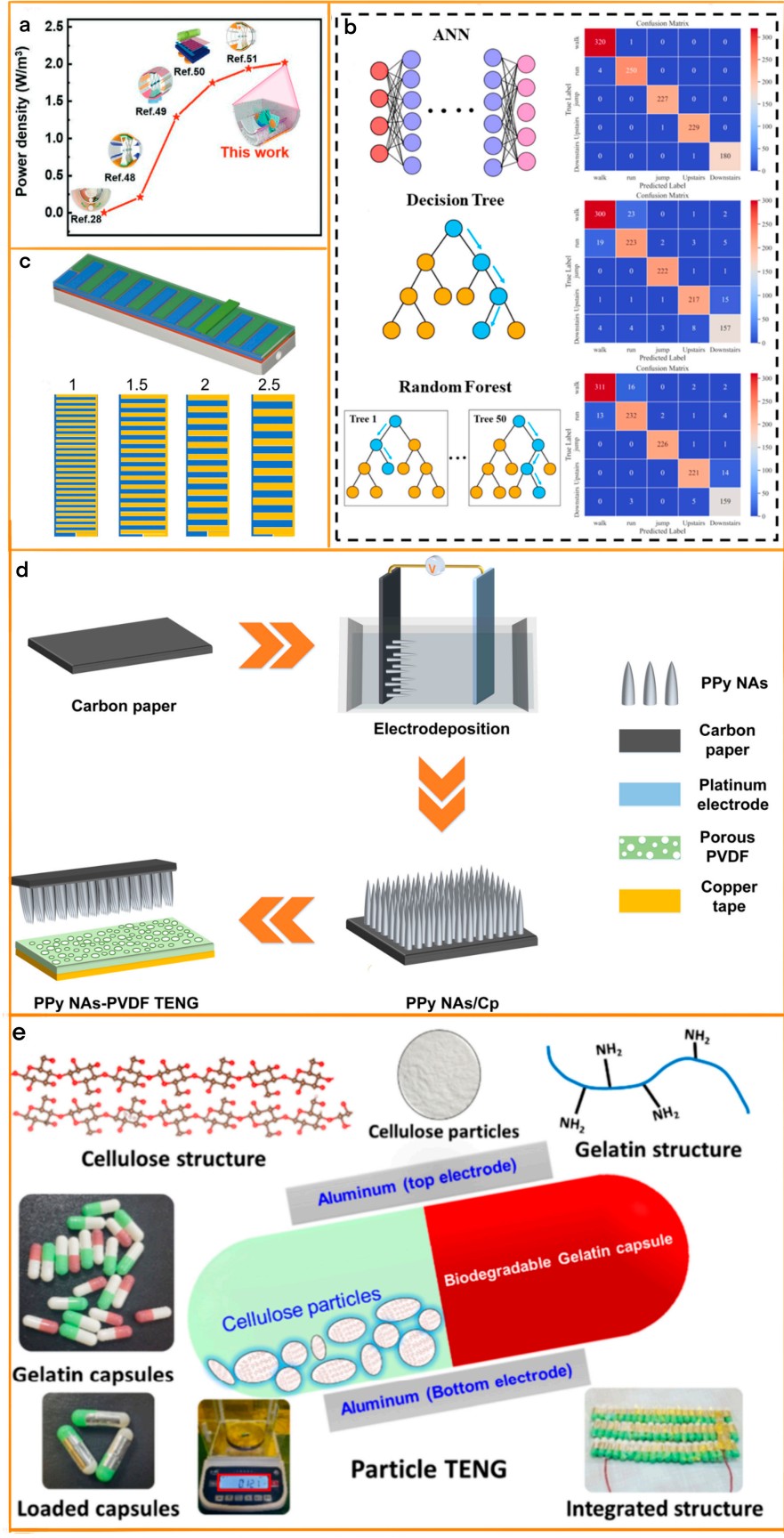

**Figure 3.** (**a**) Comparison of volume power density between this work and other similar works. 2022 John Wiley & Sons [49]. (**b**) The schematic diagram of the three models and the confusion matrix of the

gait recognition results. 2022 Elsevier [50]. (**c**) Structural diagram of the G-TENG and configuration of the varied electrode widths (1, 1.5, 2, and 2.5 mm) with a constant electrode spacing of 0.1 mm for a G-TENG angle sensor. 2023 ACS [51]. (**d**) Schematic illustration for the fabrication process of PPy-PVDF TENG. 3DPPyNAs were deposited on carbon paper by electrochemical deposition and combined with the porous PVDF film to construct PPy-PVDF TENG. 2022 ACS [52]. (**e**) The structural design of particle-based TENG. 2022 Elsevier [53].

Building upon recent strides in energy harvesting, Zhang et al.'s research marks a significant leap forward. This team has ingeniously crafted a hybrid structure combining three-dimensional polypyrrole nanoarrays with porous poly (vinylidene fluoride) films, a design that significantly bolsters the mechanical robustness and electrical yield of TENGs (Figure 3d) [52]. The core innovation of this approach resides in its strategic manipulation of frictional interaction, a pivotal factor in energy generation. By augmenting the contact surface area and enhancing the affinity for contact, these nanoarrays substantially elevate the TENG's efficacy. This amplification in performance positions the device as a potent and reliable energy harvesting mechanism, particularly suitable for use in personal electronic devices.

Moreover, Saqib et al. introduces a revolutionary approach to enhancing energy harvesting efficiencies (Figure 3e) [53]. In this new configuration, each individual particle, which originally functioned as a standalone unit in P-TENG, becomes an integral element of a larger array. This array, composed of multiple such particles, each housed in rapidly degradable gelatin capsules and utilizing cellulose-based materials, significantly amplifies the system's power generation capacity. As each particle contributes to the collective energy output, the overall system can now generate voltages and power at scales much higher than the original P-TENG's range of 15 to 85 volts and 5.488 to 70 microWatts. The array formation ensures efficient energy harvesting from all directions, thereby eliminating the limitations of traditional contact and separation methods. This modular and scalable approach not only makes the TENG highly adaptable to various applications, particularly those involving small and irregular movements, but also maintains the eco-friendly ethos of the original design by using biodegradable components. Consequently, the TENG emerges as a highly efficient, versatile, and sustainable solution in the realm of energy harvesting technologies.

## 3. Array Configuration Enhances TENG Sensitivity

### 3.1. Matrix-Designed TENG

3.1.1. Matrix Design Enhances Motion Sensing

In the ever-evolving field of health and technology, pioneering intelligent gait recognition systems have emerged on the basis of TENG as a significant innovation, offering transformative applications in identity recognition, physical training, and especially clinical medicine, where it shows promise in the early diagnosis of diseases like Parkinson's and hemiplegia [84–88]. These sensors capture intricate details of movements, forming complex data matrices which are subsequently analyzed by a data processing module before being interfaced with a PC for further scrutiny.

Wang developed a self-powered strain sensor from graphene oxide-polyacrylamide (GO-PAM) hydrogels, which also functions as a TENG to harness mechanical energy for powering up to 353 LEDs and electronic thermometers, thus demonstrating its potential in energizing electrical devices [50]. With real-time processing capabilities, these algorithms achieve high-precision recognition, showcasing an impressive accuracy rate of up to 99.5% for normal gaits and 98.2% for pathological gaits (Figure 3b). Consequently, this system stands as a beacon of progress in the gait analysis field, promising a revolutionary approach to early patient diagnosis and treatment, and offering a significant edge over other biometric identification technologies.

Dan and his team focused on refining human–robot interaction through the nuanced detection of finger movements and brought forth a groundbreaking innovation in the

domain of human–robot interaction systems [51]. At the heart of this advancement lies the stereoscopic structured triboelectric nanogenerator (SS-TENG), characterized by its distinctive matrix grating structure (Figure 3c). This central component, termed G-TENG, amplifies the device's performance, capable of precisely detecting finger movements by generating correlating triboelectric signals during the bending and holding states of the fingers. Remarkably, the device can discern bending angles with a minimum resolution of 4.1°, facilitating not only self-powered high-resolution angle recognition but also paving the way for further developments in triboelectric sensing or interactive applications. The intricate design of the SS-TENG represents a milestone in the pursuit of more intuitive and diverse control strategies in various sectors, hinting at the potential to redefine human–machine interaction techniques.

In the burgeoning field of wearable health technologies, the innovative creation of Qiu et al. has emerged as a significant breakthrough [89]. They have developed a TENG that astutely incorporates conductive polymers, particularly polyaniline (PANI), as electrodes. Merging this with commonplace fabrics and integrating it with polycaprolactone (PCL), the TENG notably enhances the compatibility between the device and its wearer, a crucial aspect in wearable technologies. The resultant TENG is distinguished by its exceptional softness, a characteristic that ensures comfort during prolonged use, an essential feature in wearable devices. This softness, combined with superior gas permeability, permits skin to breathe, thereby preventing discomfort or irritation over extended periods of wear. Additionally, the flexibility of the TENG allows it to endure various physical manipulations such as stretching, folding, and twisting, ensuring that its performance remains uncompromised in diverse usage scenarios. Furthermore, the TENG's innovative design enables the development of a calibration-free, self-powered sensor. This sensor is particularly adept at monitoring vital signs, a critical component in personal health management, ensuring continuous and accurate tracking of health metrics without the need for external power sources. This aspect greatly enhances the device's utility in remote health monitoring and in scenarios where consistent power supply is a challenge.

In recent years, the field of prosthetics has witnessed remarkable advancements and brought new hope and possibilities to amputees. Prosthetic limbs have evolved from mere functional replacements to sophisticated devices aiming to restore the natural feel and mobility that amputees once possessed. As researchers venture to overcome these hurdles, a promising innovation spearheaded by Chang and his team emerges to potentially redefine the future of prosthetic applications. This novel TENG-based tactile sensor array system aims to continually monitor the internal pressure distribution in prosthetic sockets and promises significant improvements over existing sensors. Crafted through a scalable and economical electrospinning process, the biodegradable PCL nanofiber membrane features an excellent surface area-to-volume ratio, fostering enhanced charge generation during triboelectrification processes. This TENG-based sensor demonstrates remarkable resilience, showcasing stability up to 10,000 cycles and a resistance to variations in temperature and humidity. It is envisaged that with further development and testing, the TENG-based sensor array system might become a standard component in prosthetic devices, aiding in the prevention of pressure sores and other common complications. As this technology approaches commercialization, it is hoped that it will usher in a new chapter of refined, patient-centric prosthetic solutions, contributing positively to the healthcare landscape.

### 3.1.2. Matrix Design Enhances Touch Perception

In a parallel stride of innovation, Huang and his team developed a novel paper-based touch sensor that stands as a beacon of potential in the rapidly advancing field of consumer electronics [90]. Central to this innovation is the innovative structure that incorporates embedded silver nanowire (Ag-NW) micro-probe arrays within a paper substrate, allowing the sensor to detect multiple touch inputs with heightened sensitivity, all while maintaining a slim profile of about 100 µm. A noteworthy feature of this design is the matrix or array format of the micro-probes, where each sensing unit comprises 36 independent micro-

probes with diameters less than 50 μm, meticulously arranged in a dense matrix. This arrangement is crucial in capturing the spatial distribution of touch pressure, essentially functioning as a digital extension of human skin. Remarkably, the fabrication process of this sensor involves a revolutionary double-sided laser printing technique, which is both mask-free and solvent-free, lending itself perfectly to the paper substrates used in this study. Through a streamlined "puncture and fill" process, micro-channels are created and subsequently filled with Ag-NWs, forming the embedded micro-probe array. This method showcases not only cost-effectiveness but also scalability, flexibility, and the potential for adapting to various shapes.

In the realm of addressing tactile deficiencies due to peripheral nerve damage, Shlomy et al.'s pioneering work on triboelectric nanogenerators (TENG-IT) marks a significant advancement, particularly in the context of array design enhancing touch perception [91]. The TENG-IT, with its ingeniously simple architecture, is fundamentally self-powered, biocompatible, and extensively customizable, making it a ground-breaking solution for tactile recovery. Central to its design is the use of polydimethylsiloxane (PDMS), nylon (Ny), and cellulose acetate butyrate (CAB), each selected for flexibility and biocompatibility, crucial traits for integration into human physiology. The TENG-IT operates on an array configuration, where PDMS forms the negative layer, and Ny and CAB constitute the positive counterparts. This layered array is adeptly embedded under the human skin, converting mechanical pressure into electrical potential. This potential, once generated, is deftly conveyed to healthy sensory nerves through an intricate network of cuff electrodes.

Li and his associates addressed limitations in sensitivity and crosstalk interference meticulously and delineated an innovative approach to the design and fabrication of flexible, skin-integrated TENG sensor arrays, with a concerted focus on crosstalk suppression and sensitivity augmentation (Figure 4a,b) [54]. By harnessing a synergistic amalgamation of 3D printing technology, solution-processed silver nanowire (Ag NW) electrodes, and polydimethylsiloxane (PDMS) triboelectric layers, the researchers have orchestrated the creation of an adept sensor array, boasting a remarkable sensitivity of 0.11 V/kPa and a comprehensive pressure detection range, facilitating nuanced detection of tactile nuances, spanning gentle touches to substantial pressures. This array stands as a testament to meticulous engineering, incorporating 100 independent sensing units arranged methodically within a 7.5 cm × 7.5 cm grid, a design which actively minimizes crosstalk interference, an attribute substantiated to restrict output to a minimal 10.8%, illustrating a promising trajectory in the spheres of tactile sensing and human–machine interface applications (Figure 4c,d).

### 3.2. Array Textile-Based TENG

3.2.1. Improve Motion Sensing through Array Textile Design

In the rapidly advancing field of wearable technologies, the array design in fabric-based TENGS seems to be intrinsic and the array structure serves as a critical component in enhancing the efficiency and functionality of these devices. It facilitates the fine-tuning of individual units, which is essential for optimizing performance and potentially introducing new capabilities in sensing and adaptive response to varying conditions.

In the recent study conducted by Zhang, a pioneering method has been developed to enhance the performance and multifunctionality of textile-based TENGs (t-TENGs) (Figure 5a) [56], a promising innovation in the realm of intelligent electronic textiles and biosensors. The novel brush coating method employed in this work allowed for the creation of a PDMS–CNT film on commercial silver textile, establishing a high-performing, eco-friendly, and versatile PCN-TENG system. This enhanced TENG notably excelled in energy collection and human motion detection, with the capacity to switch between modes optimized for either energy collection or precise movement identification, showcasing significant improvements in open-circuit voltage and short-circuit current compared to devices fabricated using traditional methods, indicating a promising future for self-powered smart wearables and intelligent electronic textiles.

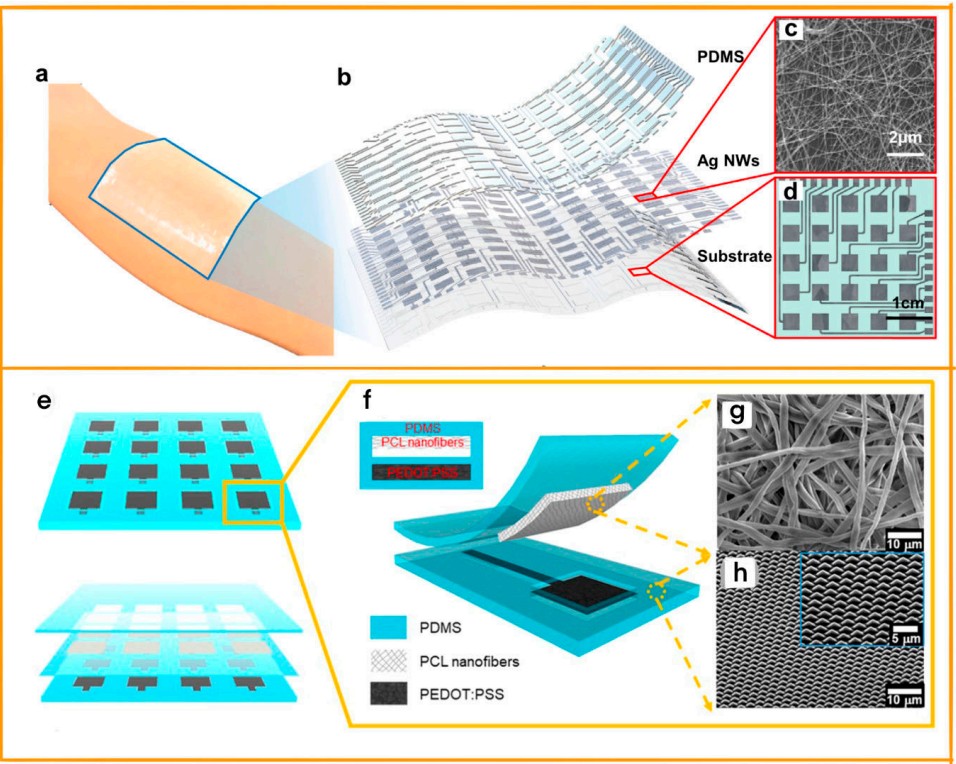

**Figure 4.** (**a**) Optical image of the e-skin mounted on the forearm of the volunteer. (**b**) Schematic illustration of the e-skin. (**c**) Scanning electron microscope (SEM) image of Ag NWs on the soft substrate. (**d**) Enlarged view of substrate with 5 × 5 pixels. 2023 Elsevier [54]. (**e**) Schematic of the tactile sensor array system. (**f**) Cross-sectional layered view of the tactile sensor based on PDMS, PCL nanofiber membrane, and PEDOT:PSS electrode. (**g**) SEM image of the PCL nanofiber membrane. (**h**) SEM image of the pyramid-like patterned PDMS layer. 2023 Elsevier [55].

Building upon the strides made in the realm of smart textiles, researchers continue to hone the intricacy and functionality of these innovative materials. In the recent study conducted by Li, a novel 3D double-electrode and machine-knittable fabric triboelectric nanogenerator (3D-FTENG) is presented as a significant advancement in wearable electronic textiles (Figure 5b,c) [57]. Developed through the utilization of coated core-spun yarn and programmable spacer fabric technologies, it integrates both positive and negative triboelectric materials along with electrode materials, where the TPU-PDMS-coated Ag-cotton core-spun yarns not only offers superior comfort and air permeability but also exhibits remarkable durability, maintaining performance even after numerous wash cycles and extensive cyclic tests. Significantly, this innovation can be incorporated into conventional garments and is able to detect a range of motions in various body parts, even foreseeing utilization in smart carpets for pedestrian flow monitoring and early warning systems.

Transitioning from Li's initiative, the research sphere witnesses another noteworthy contribution in the form of a study conducted by Dong. This study emphasizes not just energy harvesting, but also on achieving a harmonious blend of comfort and technology. Here, an innovative, stretchable, and comfortable textile-based TENG (t-TENG) was developed (Figure 5d) using Ag conductive yarns coupled with PTFE and nylon66 to showcase an array-like electronic textile [58]. This structure allows the TENG to be embedded within garments, able to harvest energy during activities such as walking or running, all the while maintaining the comfort and aesthetic of the garment. Moreover, the array design also opens up avenues in the medical field for monitoring physical parameters in real time during exercise, thus paving the way for smart garments with integrated health monitoring systems.

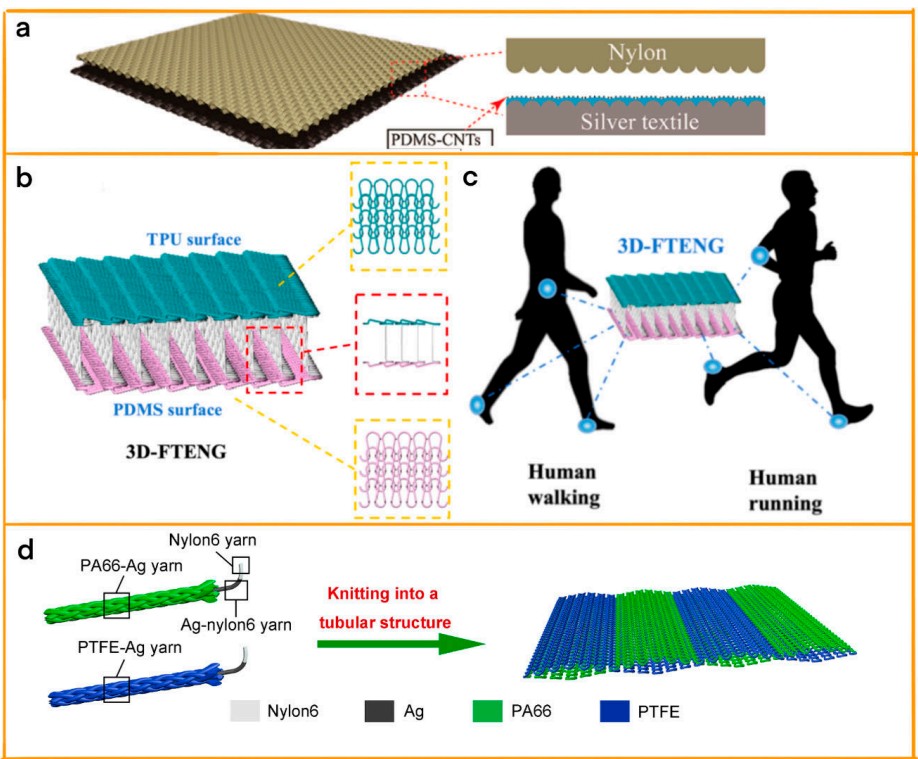

**Figure 5.** (**a**) Schematic illustration of the PCN-TENG; power generation mechanism of PCN-TENG. 2021 Elsevier [56]. (**b**) Integrated fabric morphology of the 3D-FTENG; insets are schematic views with local amplification in different directions. (**c**) Demonstration of the 3D-FTENG as a sensing textile. 2022 Elsevier [57]. (**d**) Design and fabrication of the t-TENG. 2020 Elsevier [58].

In light of the research presented, it is evident that advancements in t-TENGs are steadily progressing with a clear focus on enhancing human motion detection and energy harvesting capabilities. These works collectively highlight the potential of integrating these technologies into wearable garments, aiming for real-time monitoring of human movements and possibly assisting in healthcare applications. As these studies indicate, the field seems poised for further developments, with the goal of creating functional, durable, and comfortable smart garments that can be integrated into daily life.

### 3.2.2. Improving Disease Prevention through Array Textile Design

In recent years, the emphasis on proactive healthcare diagnostics has steadily increased and shifted from reactive medical practices to preventative strategies [92–97]. Early disease prevention, a cornerstone of this movement, seeks to detect and address potential health issues even before they manifest as noticeable symptoms, where the current strides in the development of wearable devices equipped with TENGS hold great potential to revolutionize early disease prevention strategies by facilitating continuous, real-time health monitoring.

In Dai's study, a remarkable breakthrough has been achieved in the development of wearable self-powered sensors using textile-based TENGs (t-TENGs), signifying a promising progression in the field of smart textiles [59]. The liquid alloy/silicone rubber core/shell fibers (LCFs) exhibit excellent pliability and high-resistance strain sensitivity, making them ideal for integration in wearable sensors. A prototype t-TENG with an area of $4 \times 4$ cm$^2$ demonstrated impressive electrical characteristics, such as an open-circuit voltage of 175 V and a maximum power density of 469 mW/m$^2$, and displayed versatility in detecting human motions, including walking, jogging, and running. Furthermore, the "array design" of these fibers can potentially be utilized in medical detection and serves as an invaluable tool in continuous health monitoring. The array structure could facilitate the meticulous

tracking of various physiological parameters by different units, enabling the detailed monitoring of health conditions, including minute changes in body movements, potentially aiding in early diagnosis and management of musculoskeletal disorders (Figure 6a).

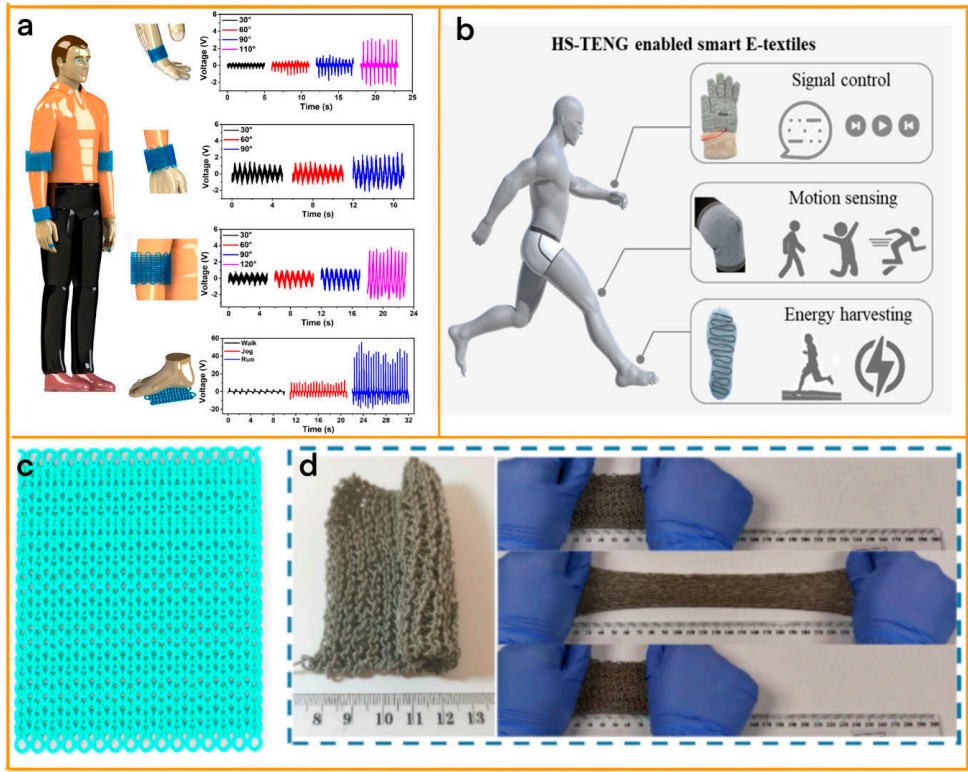

**Figure 6.** (**a**) Application of motion monitoring by the ring t-TENGs and the plain t-TENG, and the corresponding Voc under different bending angles or frequencies. 2022 Elsevier [59]. (**b**) Schematic illustration of the HS-TENG as wearable sensor integrated into various textiles, including insole for energy harvesting, kneepad for motion sensing, and glove for signal control. 2022 Elsevier [60]. (**c**) Knitted fabric structure. (**d**) Photos to demonstrate the elasticity of the knitted fabric. 2022 Elsevier [61].

As global healthcare systems are continually pressured to improve early disease prevention and monitoring, researchers are turning their attention towards the development of more sophisticated tools that can seamlessly integrate into individuals' daily lives. Central to this initiative is the advancement in smart textiles, specifically focusing on the array design of TENGS, a technology poised to significantly influence early disease detection and management.

In the research conducted by Wu, a helically structured fiber-based TENG (HS-TENG) was developed by utilizing Ti3C2Tx as the triboelectric coating [60]. This innovative structure grants the HS-TENG remarkable stretchability of up to 200% strain, and the ability to convert multiple forms of mechanical stimuli into electrical energy with impressive efficiency, allowing it to generate high electrical output even under significant deformation. This versatility facilitates its integration into electronic textiles (E-textiles), making it a promising tool for a variety of applications such as energy harvesting insoles, motion-sensing kneepads, and wireless signal controlling gloves (Figure 6b). Furthermore, this technology is primed for utilization in medical detection, potentially serving as a powerful tool for monitoring biomechanical movements closely related to joint bending.

As we delve deeper into this avenue of research, we envisage a future where technology merges harmoniously with healthcare, signifying a transformative shift towards a more proactive and informed approach to early disease prevention globally.

Cao et al. developed a versatile full-textile TENG capable of harvesting energy from various sources such as sound, human motion, and wind individually (Figure 6c,d) [61]. This device, woven with three different types of fabric, including a silicone-coated yarn, shows not only impressive electrical outputs under various mechanical impacts but also has the ability to convert low-frequency sounds into substantial electrical energy, with results that align well with theoretical predictions.

Notably, the array design of this textile-based TENG can potentially be leveraged to create innovative health monitoring systems. For instance, it could be woven into garments for the elderly to detect falls and send automatic alerts to healthcare providers or family members, potentially reducing response times in emergencies. In rehabilitation settings, the technology could facilitate precise motion analysis, aiding in the recovery from injuries or surgeries by offering real-time feedback on patients' movements, offering sensory feedback and making these devices more intuitive and functional.

## 4. Array Configuration Expands TENG Applications

### 4.1. Extreme Environmental Applications

In the contemporary world, the rapid proliferation of wearable technology and electronic devices necessitates innovations that can withstand a diverse range of environments and conditions [98–102]. A significant facet of this innovation lies in the development of devices capable of functioning under extreme stretching and in aquatic environments. Extreme stretching capabilities are quintessential in the healthcare and rehabilitation sectors, where the integration of technology into clothing or directly onto the skin requires materials with high elasticity and durability, able to maintain functionality even when subjected to significant mechanical stress [103,104]. On the other hand, the ability to function optimally in aquatic environments opens up avenues for a variety of applications, including underwater monitoring systems and wearable devices that can withstand heavy perspiration or even be used during swimming or other water-based activities. Recognizing these necessities, researchers have ventured into the exploration of TENGs and focused on the innovation of TENG arrays optimized for extreme stretching and aquatic settings.

In the pursuit of integrating TENG-based healthcare technology seamlessly with the human body for real-time health monitoring and rehabilitative support, the research spearheaded by Kim brings a significant contribution [62]. This study emphasizes the critical role of innovative array design in creating next-generation TENGS that can function effectively under extreme stretching conditions. The core of this advancement lies in a carefully engineered array, constructed from a plasticized polyvinyl chloride (PVC) gel combined with a graphene electrode (Figure 7a). This formulation, termed PGTENG, showcases a remarkable blend of stretchability and conductivity, capable of operating efficiently even when stretched up to 50%, without experiencing any decline in electrical output. Furthermore, this design excels in biocompatibility and mechanical resilience, allowing for smooth integration into wearable devices that can monitor vital signs such as heartbeat and respiration rate continuously, acting as proactive systems in healthcare. Kim's initiative thus sets a new standard in the field of biomechanical energy harnessing, offering a pathway to sophisticated wearable devices that can adapt to the dynamic demands of the human body.

While Kim's research promises a significant breakthrough in wearable technology, especially in the realms of healthcare and rehabilitation, it is essential to recognize the importance of ensuring that these devices can maintain their functionality in various environments. This brings to light another equally pressing concern: the durability and effectiveness of these technologies in high-moisture and aquatic environments. It is no longer sufficient to develop devices that only cater to one set of extreme conditions. Lv's study takes a substantial step forward, particularly addressing the challenges posed by aquatic environments and high-humidity conditions (Figure 7b) [63]. Lv introduced an interconnected TENG array that is not only flexible but also enclosed, thereby protecting the device from adverse external factors, including water and humidity, and with significantly enhanced overall output efficiency.

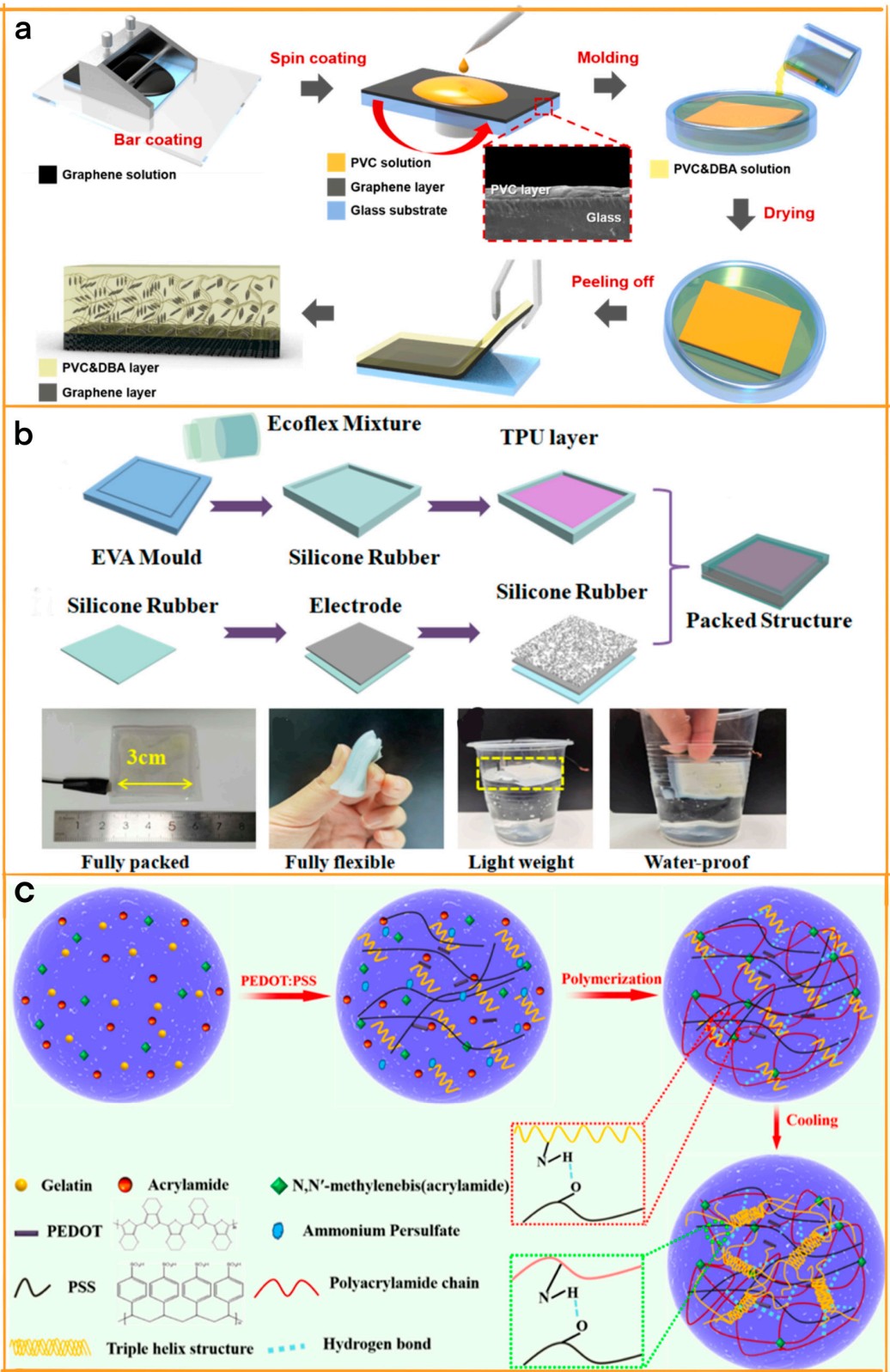

**Figure 7.** (**a**) Schematic of the fabrication process for the PVC gel/graphenebilayer. 2023 Elsevier [62]. (**b**) Schematic diagram of the fabrication process and the construction of the single enclosed TENG unit. 2021 Elsevier [63]. (**c**) Schematic illustration of synthetic procedures of MGP CHs. Elsevier 2020 [64].

Sun et al.'s innovative study explores the synergistic potential of a hybrid double network, ingeniously combining physically cross-linked gelatin and chemically cross-linked polyacrylamide (PAM), integrated with PEDOT:PSS for conductivity (Figure 7c) [59]. This unique combination yields hydrogels with exceptional properties; they are not only stretchable, conductive, and transparent, but also remarkably durable [64]. What sets these hydrogels apart is their impressive mechanical properties and self-recovery capabilities, which are primarily attributed to the intricate physical entanglements and a multitude of dynamic hydrogen bonds formed within the double networks. These characteristics are further enhanced by an array design approach, which optimizes the spatial arrangement of the composite materials, thereby maximizing their functional interaction. The embodiment of this innovative material is a transparent, wearable strain sensor that showcases extraordinary sensitivity and an ultra-wide sensing range. Its response time is notably short, and it demonstrates unparalleled durability and reproducibility, traits essential for consistent performance in real-world applications. Moreover, these hydrogels serve a dual function, acting as highly stretchable triboelectric nanogenerators (STENGs), thereby exhibiting proficient energy harvesting capabilities. This dual capacity for both strain sensing and energy harvesting, fortified by the array design, renders these hydrogels particularly promising for the development of high-performance, self-powered wearable devices and stretchable power sources.

### 4.2. Medical Applications

TENGs can serve as sensitive sensors capable of real-time monitoring and analysis, which is crucial in many healthcare and sports applications [93,105–107]. These TENGs are being employed to develop tools and devices that not only enhance data acquisition but also potentially improve quality of life and prevent injuries.

In the experiment conducted by Hao, the integral role of array design in creating the SRC-TENG is highlighted [65]. This array, developed with careful planning, is composed of a flexible and durable thermoplastic polyurethane top layer, a copper electrode, and other essential components, which work collaboratively to enable energy harvesting and real-time kinematic analysis (Figure 8a). Integrated onto a smart saddle, the array is precisely engineered to identify varying pressure areas, thereby potentially improving safety and performance in equestrian sports. Consequently, its applications can branch out into the medical field. The array's detailed design not only guarantees a quick response time but also allows for a detailed analysis of the rider's state during activity, indicating a significant step forward in the development of sports training equipment. Furthermore, it holds promise in assisting surgeons by being integrated into surgical tools, providing real-time feedback on the forces applied during surgeries, potentially improving outcomes and reducing complications.

In the recent study conducted by Kou, a significant advancement in the field of real-time mobile healthcare has been demonstrated through the development of a smart pillow [66]. Enabled by a well-designed array of flexible and breathable TENGs (FB-TENG), this approach creates a sensor system that is sensitive and noninvasive. Good durability and pressure sensitivity were also demonstrated; as shown by Figure 8b, the as-designed sensor capably facilitates tactile sensing and motion track monitoring, offering real-time monitoring of head movements during sleep, along with a supplementary feature of an early warning system to detect falls from the bed. This detailed yet efficient array design is a critical component in enhancing the sensitivity and stability of the system, setting the stage for potential applications in comprehensive healthcare management.

In recent times, the focus on developing sustainable and efficient methods for sterilization and disinfection has intensified, especially in the healthcare sector. The handling and treatment of medical waste, including synthetic urine, has emerged as a critical area of concern.

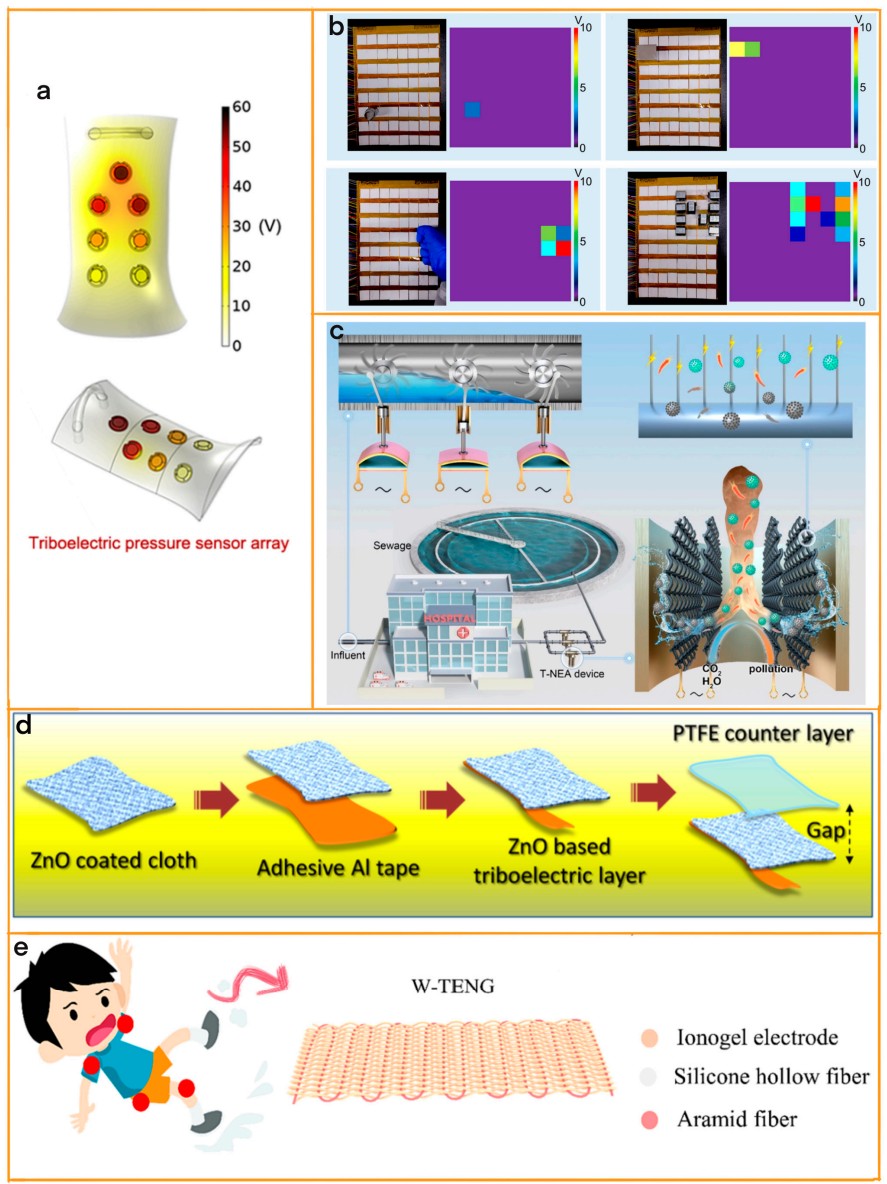

**Figure 8.** (**a**) Schematic illustration of SRC-TENG applied on the saddle and simulation diagram for the force distribution of the saddle using the COMSOL software. 2022 ACS [65]. (**b**) Application of the FB-TENG array in pattern mapping. 2022 ACS [66]. (**c**) Scheme on the treatment of T-NEA sterilization system in hospital. 2022 Elsevier [67]. (**d**) The schematic representation of the fabrication of ZnO-based active layer and the subsequent STENG [108]. (**e**) The structure and purpose of W-TENG [109].

In an inspiring work conducted by Zhang, a novel, eco-friendly T-NEA (TENG Driving Nanowires Electrode Array) system was developed to effectively disinfect synthetic urine, thereby addressing significant health and environmental concerns associated with the spread of pathogens found in urine [67]. Based on a meticulously designed TENG matrix, the system utilizes a high-voltage pulsed electric field from contact electrification and electrostatic induction, to induce irreversible electroporation damage to microorganisms, facilitating over 99.9999% bacteria inactivation, including pathogens like E. coli and S. aureus, in a span of 30 min without generating hazardous by-products (Figure 8c). Practical applications of this promising technology in the medical field include the development of portable medical disinfection devices, capable of quickly and efficiently sterilizing medical instruments without the need for chemical agents. Additionally, it could revolutionize wastewater treatment in healthcare facilities, serving as an eco-friendly solution to disinfect

wastewater, reducing the spread of waterborne pathogens and minimizing the environmental footprint of these facilities.

In this comprehensive study, Baro et al. have successfully developed a textile-based wearable sweat sensor, utilizing the principles of triboelectric nanogeneration [108]. The sensor, a single-electrode triboelectric nanogenerator (STENG), incorporates zinc oxide (ZnO) nanorods chemically grown on a textile platform, demonstrating its utility in both sweat sensing and motion detection (Figure 8d). The STENG's output variation with changing saline water concentration highlights its potential as an effective sweat sensor. This variation is attributed to the interaction between the hydrated chloride ions in saline water and the physisorbed water molecules on ZnO, enhancing the conduction band electron presence in ZnO and subsequently increasing the charge transfer between ZnO and the counter triboelectric layer. This results in a heightened output voltage. A miniaturized prototype of the STENG, approximately 1 cm in diameter, showed considerable efficacy in detecting sweat when attached to the human body, exhibiting a sensitivity of about 0.02 V/μL and a detection limit of approximately 4.8 μL. Furthermore, when integrated into a shoe insole, the device successfully detected sweat caused by foot movement. This innovative STENG also supports remote signal sensing, achieved by connecting it to a microcontroller unit and wirelessly transmitting the output to an electronic gadget. The research also explored various materials such as aluminum, nitrile, PET, and PTFE to optimize the counter triboelectric layer, finding that a PTFE-ZnO combination yielded the highest output voltage. This groundbreaking work paves the way for future advancements in wearable sensors, potentially incorporating artificial intelligence of things (AIoT), by optimizing device structure and the active layer.

Zhong et al.'s research represents a groundbreaking advancement in wearable sensor technology, particularly in the development of a triboelectric nanogenerator (TENG) textile [109]. This innovative textile, designed for power generation and active sensing, addresses common challenges in wearable sensors such as limited recognition accuracy, inflexibility, and complex manufacturing processes (Figure 8e). Central to this development is the core–shell structured conductive fiber (C-fiber), composed of ionogel and a hollow silicone tube, enabling large-scale production through a simple one-step UV curing process and offering personalized color options due to the dyeability of ionogel. Woven into a flexible fabric sensor (W-TENG), this technology excels in high-precision human behavior monitoring, maintaining its flexibility and output signal in extreme temperatures ranging from −18 to 200 °C. The W-TENG is capable of accurately detecting daily movements, types of falls, and post-fall states when attached to the human body. Notably, its application in fall detection is enhanced by employing a support vector machine (SVM) for data analysis, achieving an impressive 100% accuracy in classifying different fall categories. Additionally, the TENG can be integrated with custom Bluetooth modules for real-time fall monitoring, enhancing safety and response measures. The fabric sensor combines stretchability, heat and frost resistance, and washability with customizable size and color, making it a versatile and user-friendly option. Even after rigorous use and exposure to harsh temperatures, the sensor's electrical output remains effective, with a slight decrease in output under extreme conditions. The successful installation of W-TENG sensors on various body parts, coupled with machine learning for signal optimization, marks a significant leap in wearable technology, offering unparalleled accuracy and efficiency in human activity monitoring and fall detection.

## 5. Conclusions and Prospect

As we venture into a new period of technological advancements, TENGs with array design are emerging as a significant development in the fields of energy harvesting and real-time health monitoring. These devices, known for their modular and array-based grid-like design, hold the potential to greatly impact energy sustainability and healthcare technology. Their development signifies progress in renewable energy sectors and introduces the possibility of more accurate, real-time health monitoring systems. As we explore the

complexities of AD-TENG technology further, we discover opportunities to enhance its power generation and detection capabilities, which may contribute to a more sustainable and health-aware future.

### 5.1. Enhancing AD-TENG's Power Generation Capacity

### 5.1.1. Material Innovations

The energy collection efficiency of AD-TENGs can be significantly improved by pioneering innovations in material science, which include but are not limited to the following: (a) developing nanostructured materials, which are known to maximize surface areas available for charge transfer, potentially amplifying output power significantly; (b) exploring high-dielectric-constant materials, a pivotal step enabling an increase in charge storage capacity and facilitating a surge in energy outputs; (c) utilizing surface modification techniques such as micro/nano patterning to increase surface roughness and to optimize the contact electrification process; (d) introducing functional coatings, which is another promising avenue to modify surface potential and hence fostering more efficient energy harvesting avenues.

### 5.1.2. Structural Optimization

To escalate the energy harvesting efficiency of AD-TENGs, it is essential to prioritize meticulous structural optimization. Efforts are being concentrated on the development of hierarchical and biomimetic designs, which have shown a promising trajectory in amplifying charge generation and retention, thereby enhancing energy collection efficiency. Additionally, advanced surface modification techniques such as patterning and texturing can also facilitate an increase in the effective contact area and charge transfer capabilities, heralding significant improvements in energy harvesting efficiency.

### 5.1.3. Hybrid Energy Harvesting Systems

The integration of TENGs, including AD-TENGs, with complementary energy harvesting technologies, such as piezoelectric or solar generators, can create groundbreaking hybrid systems capable of multi-faceted energy collection. By combining the strengths of various energy harvesting mechanisms, these systems can offer enhanced efficiency and adaptability, meeting the diverse demands of various applications including wearable electronics and IoT devices, and ensuring a consistent and reliable energy supply.

### 5.2. Enhancing the AD-TENG Detection Capability

### 5.2.1. Sensitivity Augmentation

Improving the sensitivity of AD-TENG detection is key in advancing its capabilities as a health monitoring tool. Presently, novel conductive polymers and composites with higher sensitivity to target analytes are being explored, which may result in more nuanced and accurate health monitoring. Furthermore, enhancing the contact surface area or altering the surface morphology of these materials can lead to improved charge transfer and heightened sensitivity.

### 5.2.2. Ensuring Bio-Safety

In the sphere of in vivo and in vitro applications, prioritizing the bio-safety of AD-TENG devices emerges as a critical necessity. This involves not only adopting biocompatible and non-toxic materials but also fostering innovation in creating materials that supersede existing safety benchmarks. It is essential to conduct robust studies focusing on long-term stability and degradation patterns within biological settings, which is critically important for insights into potential failure modes and toxicity profiles. Concurrently, the development of advanced sterilization techniques is required, which maintain the device's functionality without compromising its stability.

### 5.2.3. Reducing Interference and Improving Accuracy

To enhance the detection capabilities of AD-TENGs further, efforts must be concentrated on reducing interference and improving sensing accuracy. Innovations such as the inclusion of built-in reference electrodes and compensation circuits or the use of selective coatings can significantly minimize environmental interference, thus improving the accuracy and reliability of these systems. Moreover, the integration of advanced signal processing techniques and machine learning algorithms can refine data analysis, leading to more accurate and robust biochemical sensing systems.

In conclusion, the future trajectory of AD-TENG technology is laden with promising opportunities and avenues for innovations in healthcare. As research deepens and technology advances, we anticipate a significant shift in the realms of energy harvesting and healthcare monitoring, steering us towards a future that harmoniously blends sustainability with advanced diagnostics technology.

### 5.3. AD-TENG in Healthcare Detection

### 5.3.1. Customization for Specific Medical Conditions

Tailoring AD-TENGs for specific medical conditions is a crucial research avenue. This involves customizing the design and operation of AD-TENGs to suit the unique requirements of different health monitoring applications, such as cardiac monitoring, glucose level detection, or neural activity tracking. Research should aim to develop AD-TENGs with specific material properties, structural designs, and operational modes that are optimized for detecting and monitoring particular health conditions.

### 5.3.2. Improving Wireless Connectivity and Remote Monitoring

As AD-TENGs are well-suited for wearable health monitoring devices, enhancing their wireless connectivity and remote monitoring capabilities is essential. Research should be directed towards integrating AD-TENGs with wireless communication technologies to enable remote health monitoring. This will allow for continuous health tracking and real-time data transmission to healthcare providers, making it possible to monitor patients in their natural environment and respond promptly to any health changes.

### 5.3.3. Developing Eco-Friendly and Sustainable Materials

Given the growing emphasis on environmental sustainability, future research should also focus on developing eco-friendly and sustainable materials for AD-TENGs. This includes exploring biodegradable and bio-compatible materials that reduce environmental impact and are safe for long-term use in healthcare applications. Sustainable material development will not only make AD-TENGs more environmentally friendly but also potentially improve their biocompatibility and effectiveness in healthcare applications.

**Author Contributions:** Z.Z.: investigation, writing—original draft and editing, conceptualization. Q.Z.: resources, supervision, conceptualization, validation. Y.W.: resources, supervision, conceptualization, validation. M.S.: resources, supervision, conceptualization, validation. X.C.: resources, supervision, conceptualization, validation. N.W.: resources, supervision, conceptualization, validation. All authors have read and agreed to the published version of the manuscript.

**Funding:** This work was funded by National Key R&D Program of China (2023YFF0612804), the National Natural Science Foundation of China (NSFC No. 51873020), and the University Basic Scientific Research Business Fee (No. FRF-MP-20-38).

**Data Availability Statement:** Data available on request from the authors.

**Conflicts of Interest:** The authors declare no conflict of interest.

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
