# Peer review of "Array-Designed Triboelectric Nanogenerator for Healthcare Diagnostics: Current Progress and Future Perspectives"

_jlpea, doi:10.3390/jlpea14010007_

Round 1

Reviewer 1 Report

Comments and Suggestions for Authors

The authors presented a novel review article on “ Array-Designed Triboelectric Nanogenerator for Healthcare Diagnostics: Current Progress and Future Perspectives. There is no review on this topic, hence it can be accepted for publication. However, Its needs significant modification to published in this journal.Below modifications need to be done and should be incorporated.

1.       Authors should mention the different review articles on TENG and after they can claim the novelty of this review. How it is different from other review articles. Authors should table of all the different review articles on TENG. This will highlight the novelty of the present review more strong.

Ex: Polymer based TENG, Textile based TENG, waste based TENG, Ferrofluid based TENG, Mxene based TENG, Crystalline porous material based TENG, Power management circuits for TENG, power enhancement strategies for TENG.

2.       Authors should also make time line of AD-TENG work from start to till date with (power density  or voltage current values) and application involved in it as table.

3.       When AD-TENG is made, authors should discuss, how current and voltage getting added up with a neat figure and how many ways this task is done in the literature.

4.       Figure 1 has copyright issue ( no citation was given)

5.       Figure 2 can be drawn , instead of taking from some reported paper

6.       Application part can extended  to another 2-3 applications. As it is review articles (ex; powering electronic devices, LEDs, etc).

7.       Spelling errors need to checked  (Ex: line 67 bisic)

Comments on the Quality of English Language

Spelling mistakes.

Author Response

I would like to express my sincere gratitude for the time and effort you have dedicated to reviewing our manuscript titled “Array-Designed Triboelectric Nanogenerator for Healthcare Diagnostics: Current Progress and Future Perspectives" Your insightful comments and suggestions have been instrumental in enhancing the quality and clarity of our work.

We have carefully considered each of your suggestions and have made the corresponding revisions to our manuscript. Please find attached a Word document that comprehensively details the changes we have implemented in response to your review. 

Reviewer 2 Report

Comments and Suggestions for Authors

The review article "Array-Designed Triboelectric Nanogenerator for Healthcare Diagnostics: Current Progress and Future Perspectives" by Zhao et al. is a detailed review on the AD-TENGs used for the conversion of mechanical to electric signals which is being widely used for biomedical application, harvest energy, monitor health and sensing diagnostics. This article can be accepted. 

Comments on the Quality of English Language

A thorough checking of grammatical error needs to be conducted.

Author Response

Thank you for your insightful and encouraging comments regarding our manuscript titled "Array-Designed Triboelectric Nanogenerator for Healthcare Diagnostics: Current Progress and Future Perspectives."

We are delighted to hear that you find our review detailed and relevant to the applications in biomedical, energy harvesting, health monitoring, and diagnostic sensing. Your positive feedback is greatly appreciated and serves as a strong motivation for our team.

We are pleased that our manuscript meets the standards for publication and look forward to seeing it contribute to the field. Please let us know if there are any further steps or revisions required for the finalization of the publication process.

Thank you once again for your constructive review and acceptance of our work.

Best regards,

Zequan Zhao

Reviewer 3 Report

Comments and Suggestions for Authors

This manuscript reports an interesting review of array-designed triboelectric nanogenerators (AD-TENGs) for healthcare detection. This review includes the working principle and design stages, applications, and perspectives. However, the authors can improve their manuscript based on the following issues:

1.-In the abstract, the term "deisgning basis" must be corrected.

2.- The introduction should add the advantages and challenges of using an array of TENGs compared to a single TENG. In addition, this section should consider the advantages and limitations of AD-TENGs compared to other types of nanogenerators such as piezoelectric, electromagnetic, thermoelectric, and hybrid.

3.- The second section should incorporate more information on the design of AD-TENGs, considering the triboelectric materials, electrodes, support structures, packaging, and signal processing. This section should add more schematic views or figures on the design phase of AD-TENGs.

4.- This manuscript should include a section on the optimization process of AD-TENGs. 

5.- The authors could incorporate more future applications of AD-TENGs for healthcare detection, considering critical discussions on their design, performance, advantages, limitations, and the Internet of Things. 

6.- The authors should include more information, discussions, and perspectives on the fabrication process, output performance, stability, reliability, and signal management of AD-TENGs. 

7.- The authors could add more recent references on triboelectric nanogenerators for healthcare monitoring devices.

Comments on the Quality of English Language

The English grammar can be enhanced.

Author Response

(The authors gave the same response as above.)

Round 2

Reviewer 3 Report

Comments and Suggestions for Authors

The authors improved their second version of manuscript based considering the reviewer's comments. This second version of manuscript can be accepted for publication in JLPEA.

Comments on the Quality of English Language

The English grammar is acceptable.